

# Technical note: an alternative approach to laboratory benchmarking of saltwater intrusion in coastal aquifers

Elena Crestani[1], Matteo Camporese[1], and Paolo Salandin[1]

[1]Department of Civil, Environmental, and Architectural Engineering, University of Padova, Padova, Italy

**Correspondence:** Elena Crestani (elena.crestani@unipd.it)

**Abstract.** Saltwater intrusion is a worldwide problem increasingly affecting coastal aquifers, due to climate changes and growing demand of freshwater for irrigation and human consumption. Therefore, research efforts on this topic have been intensified, aiming to achieve better predictions of the saltwater wedge evolution and design suitable countermeasures to limit the saltwater intrusion. Both physical and numerical models are essential for these purposes. This work presents a laboratory facility designed and built to simulate saltwater intrusion in coastal aquifers, with the overall goal of providing benchmarks for numerical models by means of different measurement techniques. The laboratory facility has been specifically designed to limit errors and provide redundant evaluation in the measurement of hydraulic heads and discharged flow rates. Moreover, the size of the facility allows us to monitor the saltwater wedge evolution by electrical resistivity tomography (ERT). A specifically designed ERT monitoring system was developed and verified by comparison with photos of the saltwater wedge collected at regular intervals during an experiment in a homogeneous porous medium. The experiment consisted of two phases: for the initial 24 h, the saltwater wedge evolved without any external forcing, while in the following 12 h, freshwater was pumped out through a channel drain, to simulate aquifer exploitation. The SUTRA code was adopted to reproduce the experimental results, by calibrating only the longitudinal and transversal dispersivities. Overall, the agreement between observed data, numerical simulations, and ERT results, albeit preliminary, demonstrates that the proposed laboratory facility can provide valuable benchmarks for future studies of seawater intrusion, even in more complex settings.

## 1 Introduction

Seawater intrusion in coastal aquifers is a worldwide problem caused, among other factors, by aquifer overexploitation related to human activities, such as irrigation and drinking water supply, and climate changes, whose main effect is the reduction of natural groundwater recharge rather than sea level rise, as recently demonstrated by Ketabchi et al. (2016). To prevent or limit the deterioration of both surface water and groundwater quality due to saltwater contamination, research studies have been developed to fully comprehend the problem and identify its fundamental parameters, as well as to evaluate possible countermeasures (e.g., Luyun et al., 2009, 2011; Kaleris and Ziogas, 2013).

Within this context, physical models represent a fundamental tool for evaluating the effectiveness of solutions to limit the salt wedge, while keeping under control system parameters, such as soil and water characteristics, as well as initial and boundary conditions. They can also provide a benchmark for the validation of numerical models. For instance, Zhang et al. (2002) carried



out for the first time an experiment to simulate a contaminant intrusion from a surface source in a coastal aquifer. Goswami and Clement (2007) carried out laboratory experiments to generate quantitative data sets to describe the transport patterns of intruding and receding salt wedges under different hydraulic gradients and with particular attention to the transient conditions. The results were then compared with numerical solutions obtained with the SEAWAT code (Guo and Langevin, 2002). Werner

et al. (2009) developed a controlled laboratory experiment to reproduce the upconing phenomenon in two dimensions, with different pumping rates and different freshwater-saltwater densities, to compare the experimental results with the analytical solutions by Dagan and Bear (1968). The results by Werner et al. (2009) were then analyzed by Jakovovic et al. (2011) with a density-dependent flow and transport model to explore the temporal development of experimental saltwater plumes. Many other studies used combinations of laboratory experiments and numerical models to study issues linked to seawater intrusion:

Chang and Clement (2012) and Chang and Clement (2013) focused on impacts of recharge fluxes variations and dynamics of groundwater flow and transport processes within the saltwater wedge, respectively; Morgan et al. (2013) reproduced the phenomenon of seawater intrusion overshoot; Mehdizadeh et al. (2015) simulated various pumping scenarios in order to verify the validity of a sharp-interface assumption; Luyun et al. (2009) and (Luyun et al., 2011) analyzed the performance of cut off walls and recharge wells as possible countermeasures to seawater intrusion.

In the previously mentioned studies, the experiments were monitored by taking regular camera images of the saltwater wedge, highlighted by adding dye to the salt water. The same technique was used by Robinson et al. (2015), who developed a method for automatic image analysis to convert the image light intensity to concentrations, and Abdoulhalik et al. (2017), in a small scale laboratory experiment to study the effects of underground barriers. On the other hand, Abdollahi-Nasab et al. (2010) were able to investigate the flushing of salt water by freshwater propagating seaward, monitoring the experiment by using only

pressure transducers and conductivity meters.

A common feature of the aforementioned experiments is that they were carried out in flumes or sandboxes of limited size, both in terms of length and width, the former being typically less than 1 m, with a few exceptions (e.g., Kuan et al., 2012; Dose et al., 2014), and the latter being usually less than 10 cm and even as small as 1 or 2 cm. On one hand, small widths imply that the wall effect, whereby the lateral walls represent preferential flow paths due to the relatively higher hydraulic conductivity

and lower dispersivity compared to the porous media (e.g., Somerton and Wood, 1988), plays a major role. On the other hand, measurements in short flumes can be significantly affected by an imperfect definition of the initial conditions or possible head fluctuations at the inland and seaward boundaries, making a correct definition of the gradient highly uncertain (e.g., Yalin, 1989).

The objective of this paper is to introduce a large laboratory facility designed and built to reproduce saltwater intrusion in

coastal aquifers. The use of a large flume allows us to limit measurements relative errors and to minimize the wall effects. Moreover, alternative techniques, such as the electrical resistivity tomography (ERT) (e.g., Binley et al., 1996; Kemna et al., 2002; Kontar and Ozorovich, 2006; De Franco et al., 2009; Rao et al., 2011; Pollock and Cirpka, 2012; Crestani et al., 2015), can be used to quantitatively monitor the evolution of the saltwater wedge, thanks to the large resistivity contrast between salt water and freshwater.

The capabilities of our laboratory facility are here demonstrated for a simple, but not simplistic, experiment of seawater in-





trusion into a homogeneous porous medium, including water pumping from a channel drain. A great care was used to obtain the best approximation of a uniform porous medium, to avoid uncontrolled heterogeneity effects. The salt wedge evolution was visually monitored by collecting photographs at regular intervals. The visual observation of the seawater intrusion was complemented by an ERT survey with a joint surface and cross-borehole configuration, specifically designed for the laboratory

flume. To give an example of the possible use of our laboratory facility for experimental benchmarking, simulations by the SUTRA code (Voss, 1984) were compared with the experimental results given by photographic evidence and the ERT survey.

## 2 Material and Methods

### 2.1 Experimental methods

#### 2.1.1 The sandbox setup

The sandbox used in this study is schematized in Figure 1 and measures 500 cm long by 30 cm wide by 60 cm high, with 3 cm thick plexiglass walls. Upstream and downstream from the sandbox, there are two tanks with volumes of 0.32 m$^3$ and 1.6 m$^3$, respectively. The upstream tank is filled with freshwater (density $\rho = 1000$ kg/m$^3$) and is continuously supplied by a small pump, providing freshwater recharge. The downstream tank, filled with salt water (density $\rho_s = 1026$ kg/m$^3$), represents the sea. Both tanks are equipped with a spillway that guarantees a constant water level, which is also continuously monitored by

ultrasonic sensors, allowing us to verify that the gradient does not change during the experiment. Red food dye is added to the salt water to easily visualize the salt wedge. Preliminary analyses were carried out on the dye to verify its ability to be used as a tracer for the salt water and they showed a good correlation between measured light absorbance and salt concentration in terms of electrical conductivity (Figure 2).

The porous medium in the sandbox is obtained by means of glass beads, as commonly adopted in previous studies (e.g.,

Goswami and Clement, 2007; Luyun et al., 2009; Chang and Clement, 2013; Robinson et al., 2015), due to the advantages given by the absence of chemical interactions between the dye and salt mixture with the porous matrix, allowing for multiple test repetitions. The beads were provided by Potter Industries Inc. and are characterized by a nominal size range equal to 400-800 $\mu$m, a median diameter $d_{50}$ of 0.6 mm, and a uniformity coefficient of $d_{60}/d_{10} \approx 1.5$, corresponding to a fairly homogeneous material. The porous medium was packed into the sandbox under moist conditions (e.g., Felt, 1959): the beads

were arranged layer by layer by compacting each 5-cm stratum with a 4 kg weight falling from a height of 20 cm.

A channel drain was added to the system, in order to simulate a freshwater withdrawal close to the coastal line, as common in real world cases. The channel drain is located 50 cm upstream from the saltwater tank and it is 30 cm long (intercepting the whole sandbox width), 1 cm wide and 10 cm deep. This depth ensures that the water table is always intercepted with or without water extraction. The channel drain is made of plexiglass and fiberglass, including a plastic mesh that prevents the glass beads

to enter into the apparatus.

Prior to the experiment, the porous medium was saturated with freshwater by very slowly increasing the water level in both tanks, to avoid the formation of air bubbles, and then keeping the water level higher than the porous medium height, i.e., 48 cm,




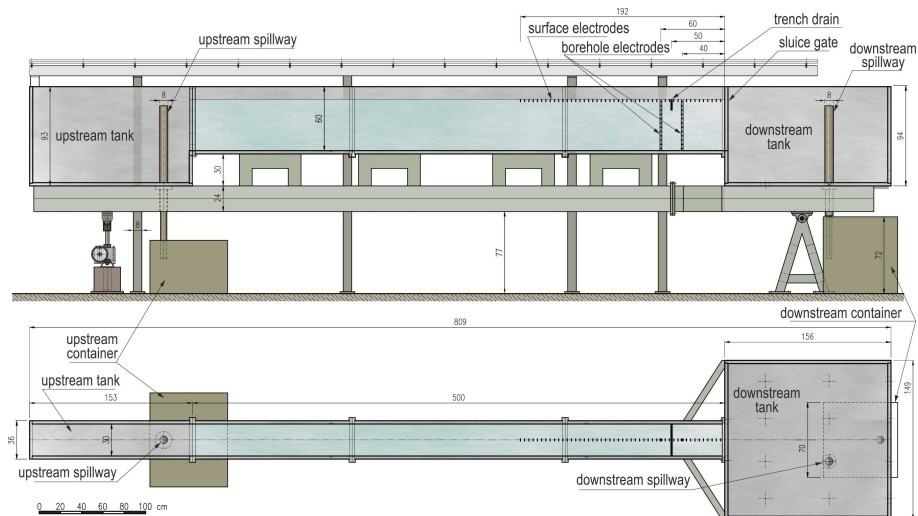

**Figure 1.** Schematic lateral (top) and plan (bottom) views of the sandbox (all distances are expressed in cm)

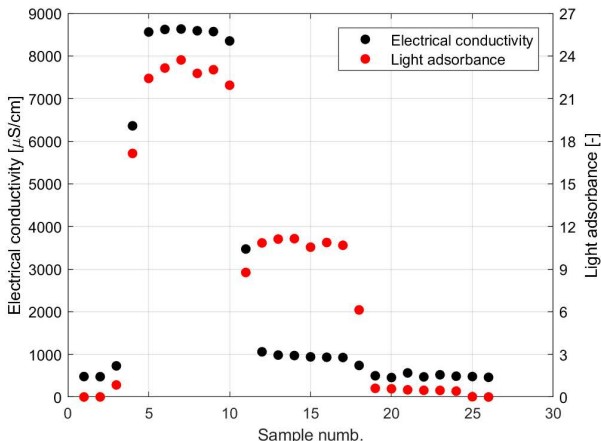

**Figure 2.** Electrical conductivity and light absorbance measured on a number of saltwater samples colored with red food dye.

for at least two weeks. When the saturation was completed, some constant-head tests were performed with different hydraulic gradients, in order to estimate the saturated hydraulic conductivity $K$. The results are shown in Figure 3, which reports, for the various hydraulic gradients, the time series of $K$ values obtained with the Dupuit formula:

$$Q/B = K(H_{up}^2 - H_{dw}^2)/(2L), \tag{1}$$

where $Q$ is the discharged flow rate, $B$ is the width of the sandbox, $H_{up}$ and $H_{dw}$ are the upstream and the downstream heads,
5   respectively, measured from the aquifer bottom, and $L$ is the sandbox length.

Upon reaching steady state, typically after a few hours, all the tests converged to $K$ values of approximately $1.3 \times 10^{-3}$ m/s.





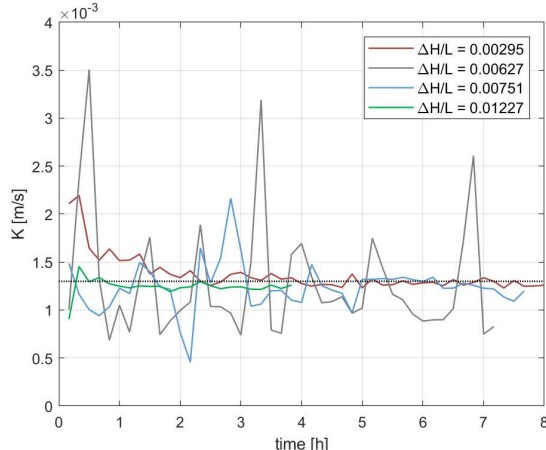

**Figure 3.** Results of the constant-head tests to determine the saturated hydraulic conductivity $K$.

The tests were also useful to improve the model facilities, as these preliminary experiments highlighted the need to use cylindrical blades for the ridge of the spillways to minimize the disturbances related to surface tension, which is evident in the $\Delta H/L = 0.00627$ case of Figure 3.

### 2.1.2 Monitoring equipment

Acoustic probes Pepperl+Fuchs UC500-30GM-IUR2-V15, with data collection frequency of 30 Hz, are used not only to monitor the water level in the upstream and downstream tanks, but also to measure the discharged flow rate. The experiment is characterized by a hydraulic gradient of about 0.38 %, a small value that can be representative of real field conditions (e.g., Antonellini et al., 2008; Yechieli et al., 2010). This can be obtained here thanks to the large size of the flume and thus the relatively limited errors in the hydraulic head measurements. As a result, the corresponding discharged flow rate is also small,

of the order of a few liters per hour. Therefore, special attention must be paid to obtain reliable measurements of the flow rate. To this aim, the water discharged from the spillway in the downstream tank is collected in a calibrated container placed under the tank and the flow rate is derived from measurements of the water level variation recorded every 10 min. The system setup (including the upstream tank and spillway characteristics) is designed to ensure that the required small flow rate can be assumed as constant throughout the experiment.

The expected freshwater inflow is small compared to the capacity of the downstream tank (about 5% of its volume), in order to prevent a significant dilution of the salt water. Nonetheless, to monitor the saltwater concentration in the downstream tank, water electrical conductivity is measured every 20 min by a portable conductivity meter WTW Multiline P4.

The saltwater wedge evolution is monitored with two different methods: i) photographs of the red-colored wedge are collected at a 5-minute interval with a Nikon D7000 camera and ii) electrical resistivity tomography (ERT) is used to derive images of

electrical resistivity distributions every 20 min.





The ERT survey is carried out with a joint surface and cross-borehole configuration, specifically designed for the laboratory flume. Forty-eight gold-plated electrodes of 1.5 cm length and diameter of a few millimeters are placed on the surface. The pins are inserted into the porous medium along the central axis of the sandbox, at regular 4-cm intervals, corresponding to a total length of 1.9 m toward the downstream end of the flume, which is the part crossed by the saltwater wedge. As for the

cross-borehole part of the ERT configuration, two slotted PVC tubes of 1.7 cm diameter were driven into the porous medium at distances of 40 and 60 cm from the downstream tank along the central axis of the sandbox. The equivalent hydraulic conductivity of the slotted wells is larger than the one of the porous medium, as demonstrated by preliminary infiltration tests (not shown here). On the other hand, the size of the slots is small enough to prevent the glass beads from entering the boreholes. Twelve electrodes are placed in each borehole, by means of a cylindrical rod in which the electrodes are embedded at regular

4-cm intervals. These electrodes are made of stainless steel, to avoid potential corrosion due to the highly saline solution. Raw electrical data are collected every 20 min by an IRIS Syscal Pro 72 Switch resistivity meter. Each data acquisition takes about 7 min, a time small enough compared to the dynamics of the salt wedge to avoid "blurred" electrical images, and consists of 3315 different combinations of surface-only, cross-hole, and surface-to-hole quadrupoles, in both pole-dipole and dipole-dipole configurations. Raw data are then post-processed in inverse mode with the ErtLab software (Geostudi Astier S.r.l.,

2015) to obtain spatial distributions of bulk electrical resistivity, i.e., the inverse of conductivity. The inversions are based on a standard smoothness constrained least squared algorithm (Geostudi Astier S.r.l., 2015).

### 2.1.3   Experiment initialization

Due to the large size of the seaward tank, it is not possible to fill it instantaneously by a tracer slug, as done in previous studies (e.g., Goswami and Clement, 2007). Therefore, our initialization procedure is more complex. Initially, a flow of freshwater only

is driven by a gradient of 0.38 % ±0.01%, resulting from water levels of 42.6 cm and 40.7 cm in the upstream and downstream tanks, respectively. This is maintained for 24 h to ensure the achievement of steady state hydraulic flow conditions and to check once more the hydraulic conductivity and discharged flow rate values before the experiment. Then, the closure of a sluice gate (Figure 1), installed between the sandbox and the downstream tank, allows the replacement of freshwater with salt water, up to the original level of 40.7 cm. These operations last about 20 min, during which the freshwater flow stops, and the water in

the sandbox levels off at 42.6 cm. Then, the sluice gate is lifted and the saltwater wedge is allowed to infiltrate into the porous medium, effectively marking the start of the saltwater intrusion experiment.

### 2.2   Numerical modeling

The saltwater intrusion experiment was reproduced by using SUTRA (Saturated-Unsaturated TRAnsport) (Voss, 1984), a well-established numerical model that can simulate saturated-unsaturated and density-dependent groundwater flow based on

quadrilateral finite elements.

The computational domain, representing the sandbox, was discretized by a total of 237500 elements, corresponding to 1250 by 190 quadrilaterals along the horizontal and vertical directions, respectively. Two different element sizes were used (0.25 cm × 1 cm and 0.25 cm × 0.25 cm), with the smaller elements closer to the downstream tank, where the saltwater wedge mostly





evolves.

The simulation time step was 1 min and boundary conditions were assigned as follows. No flow was assigned to the nodes at the top and at the bottom of the domain, as well as the nodes along the lateral boundaries located above the tank water levels. To the lateral nodes located below the upstream tank water level, hydrostatic pressure was assigned and the salt concentration

was fixed to zero. To the lateral nodes located below the downstream tank water level, the boundary conditions were assigned to reflect the conditions observed during the experiment. Although the height of the saltwater wedge at the seaward boundary reached a stable level of approximately 36.7 cm in about ten hours after lifting the sluice gate, this transient phase was neglected in the numerical model. For simplicity, freshwater hydrostatic pressure was imposed to the nodes from 36.7 to 40.7 cm, while saltwater hydrostatic pressure was assigned to the nodes below. Accordingly, salt concentration was set to 46.34 g/l

in the nodes from 0 to 36.7 cm and to zero in the upper nodes. More detailed and accurate seaward boundary conditions could be taken into account by imposing a global flux condition on both freshwater and saltwater fluxes (e.g., Abarca et al., 2007; Goswami and Clement, 2007), but this would go beyond the goals of this note, which is not primarily focused on modeling.

The channel drain is simulated by means of four columns of 40 elements characterized by a hydraulic conductivity value three orders of magnitude larger than the porous medium. The extraced flow rate is distributed in the lower eight nodes of the central

column. The initial conditions for the simulation correspond to the experimental system state resulting from the sluice gate operations carried out to fill the downstream tank with salt water and that temporarily stopped the freshwater flow.

The porous medium was assumed homogeneous, with $K$ equal to $1.3 \times 10^{-3}$ m/s (see Figure 3) and porosity $n = 0.367$, as derived from geotechnical tests. The water density was also assigned on the basis of measured values, equal to 1000 and 1026 kg/m$^3$ for freshwater and salt water, respectively. The unsaturated zone in the experiment is not relevant, as its mean thickness

in 6 cm, compared to a total thickness of the porous medium of 48 cm. Also, the flow is basically horizontal, because there are no sources of recharge from the surface. Therefore, the unsaturated zone is characterized in the model by a retention curve describing a generic fine sand.

The longitudinal ($\alpha_L$) and transversal ($\alpha_T$) dispersivity coefficients were the only parameters calibrated by fitting the modeled salt wedge shape and toe position to the observed data. The resulting values ($\alpha_L = 0.001$ m and $\alpha_T = 0.0001$ m) are consistent

with the guidelines based on the mesh-based Péclet number, as indicated by Voss and Provost (2010), and with values found in the literature (e.g., Oswald and Kinzelbach, 2004; Goswami and Clement, 2007; Abarca and Clement, 2009; Walther et al., 2012; Mehdizadeh et al., 2015).

## 3   Results

### 3.1   Experiment phase with no external forcing

The first part of the experiment was carried out by letting the saltwater wedge evolve in the sandbox for 24 h without pumping water from the channel drain. The continuous monitoring system allowed us to record the temporal behavior of the main physical quantities that describe the evolution of the phenomenon. The analysis of the water levels in the upstream and downstream





**Table 1.** Summary of the variables and parameters of the physical and numerical models.

| Variable or parameter | Symbol | Value | Type |
|---|---|---|---|
| Porous Medium thickness | $B$ | 0.48 m | measured |
| Upstream freshwater head | $H_{up}$ | 0.426 m | measured |
| Downstream saltwater head | $H_{dw}$ | 0.407 m | measured |
| Freshwater density | $\rho_f$ | 1000 kg/m$^3$ | measured |
| Saltwater density | $\rho_s$ | 1026 kg/m$^3$ | measured |
| Permeability | $k = K\mu/\gamma$ | $1.30\times10^{-10}$ m$^2$ | measured |
| Longitudinal dispersivity | $\alpha_L$ | 0.001 m | calibrated |
| Transversal dispersivity | $\alpha_T$ | 0.0001 m | calibrated |
| Porosity | $n$ | 0.367 | measured |
| Discharged flow rate (no pumping) | $Q_{disch}$ | 1.6 l/h | measured |
| Discharged flow rate (pumping) | $Q_{dp}$ | 0.3 l/h | measured |
| Pumped flow rate | $Q_{chan}$ | 1.3 l/h | measured |

tanks confirms that the hydraulic gradient remained constant during the experiment. The discharged flow rate ($Q_{disch}$), i.e., the difference between exiting freshwater ($Q_{fresh}$) and entering salt water ($Q_{salt}$) across the seaward boundary, is inferred from the volumetric measurements of water exiting through the spillway of the downstream tank and its value was estimated at about 1.6 l/h. The mean measured values of the experimental discharged flow rate ($Q_{disch\,exp}$) is reported in Figure 4, with the errors of the measures, represented through a grey band of $\pm2\sigma_m$ around the mean value, $\sigma_m$ being the standard deviation of the measures. Figure 4 also reports the simulated and the experimental flow rate pumped by the channel drain ($Q_{chan}$ and $Q_{chan\,exp}$, respectively). The green band has an height of $\pm2$ the standard deviation of the measured pumped flow rate, around the mean value. Figure 4 also shown the simulated $Q_{in}$, i.e., the sum of $Q_{disch}$ and $Q_{chan}$.

The electrical conductivity measured in the downstream tank was constant (average value of 68.45 mS/cm and standard deviation of 0.18 mS/cm), demonstrating that no significant salt dilution occurred, being the integral volume of freshwater released during the experiment negligible compared to the saltwater tank capacity.

The saltwater wedge at 24 h since the opening of the sluice gate can be visualized in Figure 5 as photographed by the camera, in terms of electrical resistivity distribution derived by the ERT inversion, and as simulated by SUTRA. The qualitative correspondence between the photograph, the model simulation, and the electrical resistivity distribution is quite good, despite the preliminary character of the ERT inversions. The main differences concern the transition zone between freshwater and salt water: while the ERT image and the numerical model show a relatively thin, but evident, mixing zone, the photo exhibits a very sharp interface, without visible evidence of a mixing zone. This may be due the fact that the camera is only able to capture what is happening at the contact between the fluid and the plexiglass or to the weak correlation between light absorbance and salt concentrations in the intermediate range (Figure 2), preventing the visual identification of a mixing zone. On the other hand, the presence of an evident diffusive zone in the ERT image may be due to overdispersion that typically affects ERT inversion





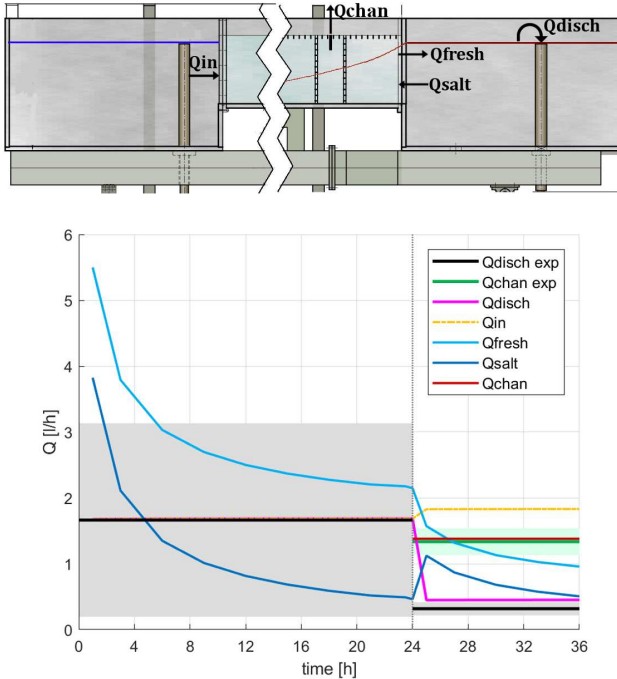

**Figure 4.** Top) Schematization of the flow partitioning in the sandbox and (bottom) the simulated and measured flow rates as represented in the scheme.

products (e.g., Binley and Kemna, 2005; Mosegaard, 2011). The ERT image also exhibits irregular features in correspondence of the boreholes, especially in the transition zone. These inversion artifacts are likely produced by interferences in the electrical field, caused by an imperfect design of the PVC rods used to connect the borehole electrodes.

Despite the significant dispersion of the electrical resistivity values, the correlation with concentration values is evident. We

5  show this by analyzing the data pairs of simulated concentration and electrical conductivity (i.e., the reciprocal of resistivity) values occupying the same location in the downstream 1.95 m portion of the sandbox, i.e., the area crossed by the saltwater wedge. The results of this analysis are shown as a density plot in Figure 6, where the color of each cell represents the count of data pairs falling into the cell itself. As the grid used for computing the concentrations is finer ($0.0025 \times 0.0025$ m$^2$) than the one used for the electrical conductivities ($0.02 \times 0.02$ m$^2$), the concentration values were averaged over the cells of the electrical

10  model grid. Although the large majority of data pairs are concentrated around zero and maximum concentration values, Figure 6 also highlights some correlation for the concentration values in the transition zone (approximately in the range 0.01–0.03 kg$_{solute}$/kg$_{water}$). The linear regression line, computed considering all the data pairs, shows a weak, albeit statistically significant, positive correlation (coefficient of determination, $R^2$, equal to 0.16). We acknowledge that the true relationship between concentration and bulk electrical conductivity may be more complex, being specific to the experimental conditions and design,

15  but a more advanced analysis is beyond the scope of this note. Nevertheless, the results clearly show the potential benefits





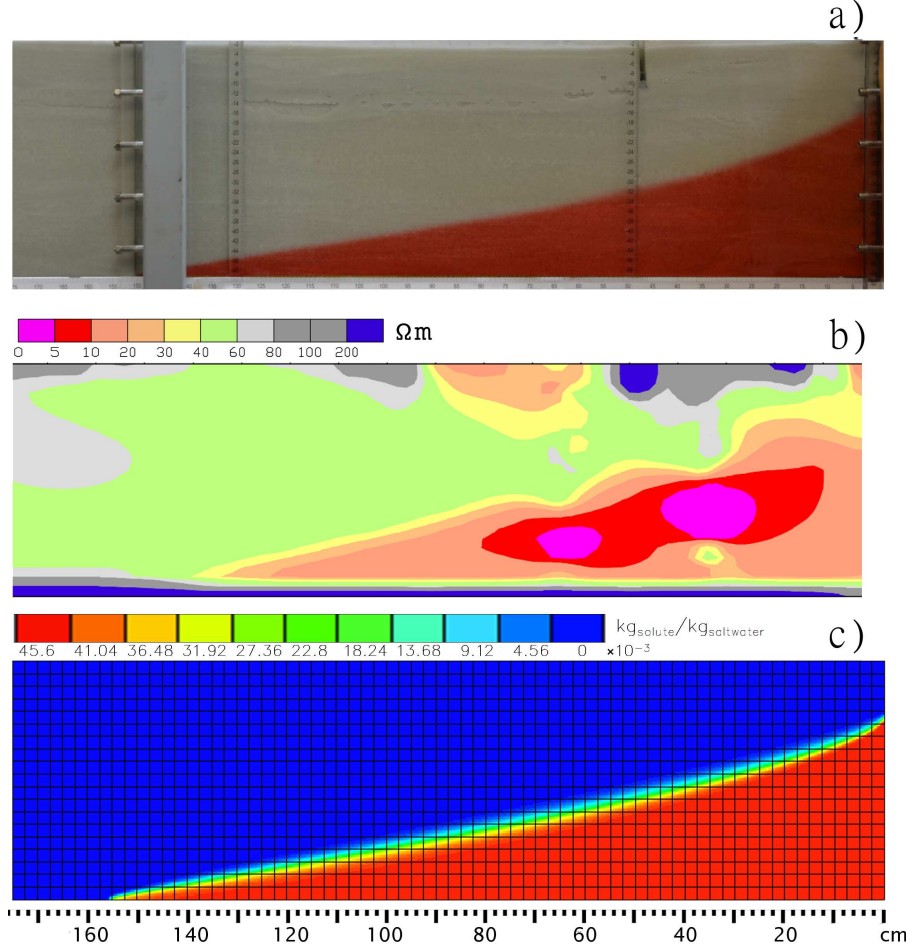

**Figure 5.** Illustration of the saltwater wedge after 24 h from the beginning of the experiment: a) photograph, b) electrical resistivity distribution as computed by the ERT inversion, and c) salt concentration as simulated by SUTRA.

associated to ERT monitoring.

Figure 7 reports the distance of the saltwater wedge toe, $x(t)$, from the downstream tank, as simulated by SUTRA and as derived by image analysis of the photographs, together with the advancement velocity of the saltwater wedge toe, $v(t)$:

$$v(t) = \frac{x(t_i) - x(t_{i-1})}{t_i - t_{i-1}}, \tag{2}$$

with $t_i - t_{i-1} = 1$ h. The toe moved faster at the beginning of the experiment, then it gradually slowed down, reaching an approximately constant velocity after about 12 h. Experimental and numerical results are in good agreement, except for a slight overestimation of the toe position in the model simulation.





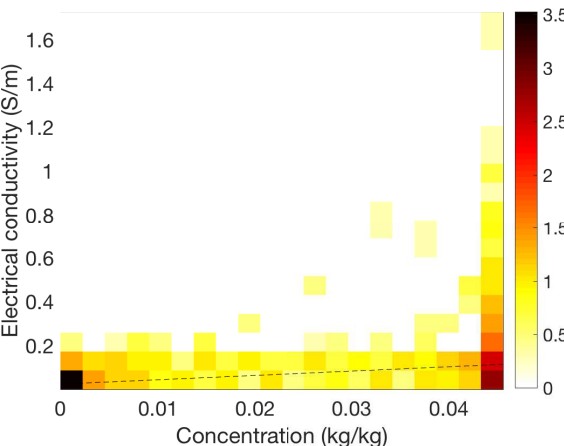

**Figure 6.** Density plot of inverted electrical conductivity as a function of simulated salt concentration after 24 h from the beginning of the experiment. The color bar indicates the $\log_{10}$ data count in each cell. The black line is the regression line.

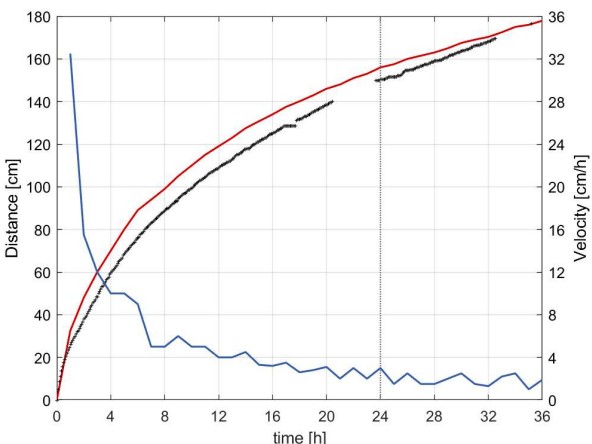

**Figure 7.** Position of the saltwater wedge toe measured from the downstream tank, during the physical experiment (black crosses) as revealed from image analysis of the photographs, and simulated by SUTRA (red line). The blue line represents the advancement velocity of the toe, as computed from the model simulation. The vertical dotted line marks the end of the first part of the experiment (without pumping) and the beginning of the second (with pumping)

## 3.2 Experiment phase with pumping forcing

After 24 h from the beginning of the experiment, water pumping through the channel drain was activated, causing a reduction of the freshwater flow downstream of the channel drain. This is a realistic reproduction of a common situation in coastal aquifers exploited for drinking water supply and irrigation.



Figure 8 shows a photograph of the sandbox after 9 h of pumping, together with the corresponding snapshots of the SUTRA model simulation and electrical resistivity distribution inferred by the ERT data. A slight upconing effect due to the pumping is clearly visible in the ERT image and in the model simulation, while just a hint can be deduced from the photograph. This could be due again to the lack of a visible transition zone in the photograph, as discussed in the previous section, and highlights the need to combine different measurements techniques as proposed here.

The density plot (Figure 9), computed as in the previous case, shows a stronger correlation between simulated salt concentrations and inverted electrical conductivities compared to that at 24 h. At 32 h, the saltwater wedge covers a larger portion of the sandbox than at 24 h; therefore, the area investigated by the ERT system is larger and characterized by higher electrical conductivities, so that the electrical inversion errors are smaller than in the previous analysis. The linear regression shows in this case a more clear ($R^2 = 0.31$) and still statistically significant correlation.

As shown in Figure 7, the saltwater wedge toe velocity did not vary significantly after the pumping started (from 24 h to 36 h), as flow conditions were not affected by the pumping upstream of the channel drain. Both numerical results and ERT evidence clearly show how water pumping caused a local increase of the transition zone thickness, especially at a distance of 40 cm, i.e., the nearest distance downstream from the channel drain. This effect gradually reduces as the distance from the downstream tank increases. These findings are consistent with numerical results reported in the literature (Darvini and Salandin, 2002).

## 4  Summary and Conclusions

In this paper we presented a laboratory facility designed to carry out experiments of seawater intrusion in coastal aquifers to serve as potential benchmarks for numerical models. As an example, we carried out an experiment in a homogeneous porous medium considering the salt wedge evolution deriving from an instantaneous application of a constant seaward boundary condition in a free surface aquifer characterized by a constant gradient. Freshwater pumping from a surface channel drain crossing the entire with of the aquifer was also considered. The experimental findings were then reproduced with the SUTRA code. In the numerical model, only the longitudinal and transversal dispersivities were adjusted to fit the shape and advancement of the salt wedge toe, while all the other model parameters were assigned based on the experimental data.

The width and length of our laboratory facility, i.e., a specifically designed sandbox, are, to the best of our knowledge, larger than most of the others used for the same purpose, allowing for the use of electrical resistivity tomography (ERT), with a joint surface and cross-borehole configuration, as a monitoring tool, along with visualization of the saltwater wedge by photographs. The flume length of 500 cm allows the salt wedge to span hundred of centimeters during long experiments, making the effects of possible head fluctuations at the inland and seaward boundaries and uncertaintites on the initial conditions negligible. Furthermore, our experimental conditions are highly controlled and the main factors affecting the saltwater intrusion can be monitored for the entire experiment duration. The hydraulic gradient, the discharged flow rate and the salt concentration in the downstream tank are continuously measured, while the hydraulic conductivity can be determined through constant-head tests, resulting in an experimental system with redundant measurements and a careful definition of their uncertainty.

The approach we suggest to monitor the saltwater wedge evolution is a combination of two complementary methods. The





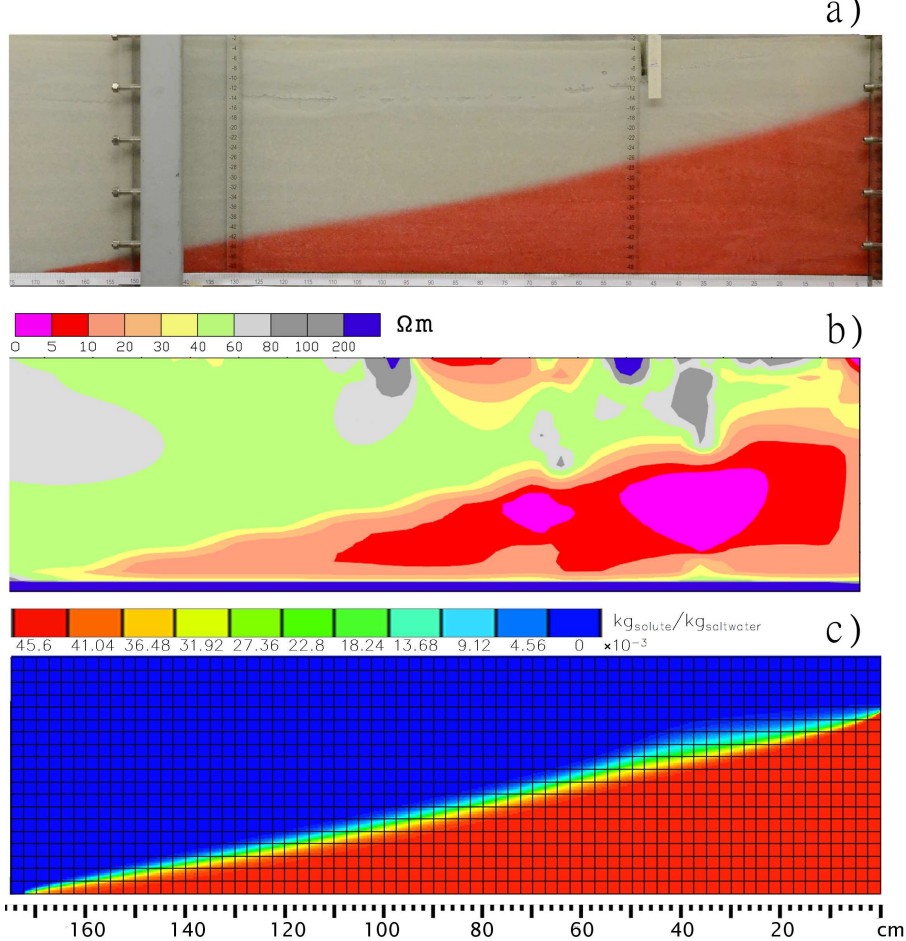

**Figure 8.** Illustration of the saltwater wedge after 32 h from the beginning of the experiment: a) photograph, b) electrical resistivity distribution as computed by the ERT inversion, and c) salt concentration as simulated by SUTRA.

visual observation of the saltwater intrusion, by means of photographs taken at regular time intervals, may be affected by errors related to wall effect phenomena, parallax and light refraction, as well as the weak correlation between light absorbance and salt concentration that makes it difficult to identify the transition zone. In order to overcome these issues and to achieve a more comprehensive description of the process, a properly designed ERT system can be used to monitor the evolution of the saltwater wedge. The ERT inversions showed here are preliminary and affected by some limitations, but the development of high-performance inversion methods is outside the scope of this work. Nonetheless, despite a significant dispersion typical of the ERT inversions, we found statistically significant correlations between the salt concentrations simulated by SUTRA and inverted electrical conductivity values, which also showed evidence of the presence of a transition zone. Overall, the ERT results proved to be effective in highlighting the effects of the pumping, especially on the transition zone dynamics.





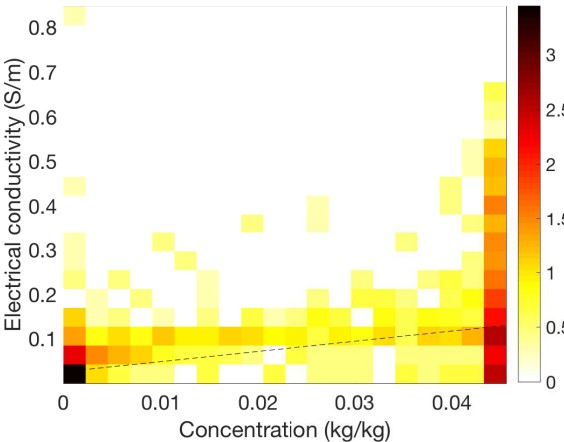

**Figure 9.** Density plot of inverted electrical conductivity as a function of simulated salt concentration after 32 h from the beginning of the experiment. The color bar indicates the $\log_{10}$ data count in each cell. The black line is the regression line.

We recall here that the numerical simulations were set up to reproduce exactly the laboratory experiment in all its phases, including the preliminary aquifer saturation and downstream tank fill-up. The simulation results were in very good agreement with the experimental observations, in terms of both saltwater wedge evolution and total flow discharge. Although the longitudinal and transversal dispersivities had to be calibrated, the comparison between observed and simulated data confirm that our

5    experimental facility can be effectively used to develop benchmarks for the validation of density-dependent flow and transport numerical models.

Future directions of the research will include improvements and developments. For instance, we plan to remove the interferences in the electrical field caused by the PVC rods in which the borehole electrodes are embedded and some metal components of the sandbox structure, such as the vertical gate at the seaward boundary, which probably alter the ERT measurements. Given

10   the successful testing of the proposed experimental facility and investigation techniques in a simple homogeneous case, we plan to investigate the effects and performance of cut-off walls as countermeasures to the seawater intrusion, also in a more realistic setting with heterogeneous porous media.

*Data availability.* All the data are available upon request to the corresponding author.

*Competing interests.* The authors declare that they have no competing interests.



*Acknowledgements.* This study was funded by the PRIN 2010-11 project 20104J2Y8M_005 ("Hydroelectric energy by osmosis in coastal areas") and the University of Padova project CPDR135701/13 ("Laboratory investigation of a river intake affected by coupled surface-subsurface saltwater intrusion"). We gratefully acknowledge Erika Bertorelle and Federico Costantini for helping with the experiments, in fulfillment of their MSc thesis, as well as Enrica Belluco, who was instrumental in the setup of the physical model. We also thank Barbara

5  Chiozzotto and Marco Giada for the collection and analysis of the ERT data.





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
