# Peer review of "Technical note: an alternative approach to laboratory benchmarking of saltwater intrusion in coastal aquifers"

_Hydrology and Earth System Sciences, 2019_

## Referee Comment (RC1) · Anonymous Referee #1 · 6 May 2019

The manuscript by Crestani et al. attempts to develop a benchmark case for saltwater intrusion using various methods. Unfortunately, I find that the research does not achieve this aim, in that the results of the experiment do not extend significantly beyond previous experimental results showing seawater extent in a laboratory coastal aquifer. I also challenge the suggestion that the ERT results help to characterise the salinity distribution - and rather, the ERT results are far less reliable than the dye tracer distribution obtained from photographs.

The following are specific issues.

Figure 2 – I don't understand how this diagram shows "good correlation between mea-

[Figure]

sured light absorbance and salt concentration". "some constant-head tests" doesn't describe what was done. The parameters of those tests should be provided. "H" should be in italics. The values of K from individual tests should be reported. Where it states "all the tests converged to K values of", what does this mean? Converged in time? As more tests were undertaken, the average value converged on 0.0013 m/s? Figure 3 is confusing. How are time-variant K values obtained? If steady-state conditions only arise after a few hours, why are values reported before then? Was a steady-state equation used for non-steady-state conditions? If so, this is wrong and should be re-thought. It is not useful to present results of a steady-state equation applied to non-steady-state results. Figure 3 – please use correct italicization for variables in the legend and axes labels. The statement "use cylindrical blades for the ridge of the spillways to minimize the disturbances related to surface tension" hold no meaning to me. Please explain in terms that are meaningful, clear and unambiguous, or remove this statement. Figure 3 – the case 0.00627 is clearly erroneous. Was it included in the arriving at the value of 0.0013 m/s for K? Normally, a result such as this would be removed on the basis of experimental erro In 2.1.2 "TM" or $^{\circledR}$ is probably needed when describing trademark and registered names. In 2.1.2, I suspect that the acoustic probes were used only to obtain the water level, but then the flow was determined from a rating curve determined for the outflow. True? If so, please explain this properly. Otherwise, how an acoustic sensor was used for both flow and water level is hard to understand. 2.1.2 – 3.8% rather than 3.8 %. 2.1.2 – "obtained here" should be "used here". The gradient wasn't "obtained" as much as it was set by the researchers. 2.1.2 – "discharged flow rate" should be "discharge flow rate" 2.1.2 – It is not correct to assume that the flow rate was constant during an experiment in which the interface was moving inland. Changes to the interface location will change the flow rate, because seawater is entering the aquifer changing the storage of freshwater. 2.1.2 – This doesn't make sense to me: "The system setup (including the upstream tank and spillway characteristics) is designed to ensure that the required small flow rate can be assumed as constant throughout the experiment." The upstream tank and the spillway do not create constant flow. The
experiment controls that – including movement of the interface and the heads as both ends, plus the initial conditions, and the time-length of the experiment. 2.1.2 – What was done to maintain the salinity of the downstream tank? It is not enough to presume that the flows were small and therefore the downstream boundary shouldn't change by much. Monitoring it is useful to a degree, but previous experiments of seawater intrusion take steps to maintain the EC of the sea boundary. 2.1.2 – Suggest past tense for verbs used in sentences that describe actions taken in the past – e.g., "system was designed", "conductivity was measured" and so on. 2.2 "237,500" – please use delimiters in large numbers. 2.2 – This explanation does not help the reader know what downstream boundary conditions were used "To the lateral nodes located below the downstream tank water level, the boundary conditions were assigned to reflect the conditions observed during the experiment." There are several options in SUTRA for assigning boundary conditions to represent the sea. The explanation in 2.2 describes a deficient manner of simulating the sea boundary condition. SUTRA can simulate flow-in-flow-out boundary conditions, whereby the direction of flow is accounted for. This should have been done. Rather, the authors have used the outdated approach of predetermining the freshwater outflow depth instead of letting the model calculate it. In 2.2, I disagree with the approach of adopting freshwater hydrostatic pressure from 36.7 to 40.7, based on the location of the interface. This is wrong. The seawater boundary condition applies the same pressure to the aquifer regardless of the salinity in the aquifer. If freshwater is flowing outwards, the head of the sea should not change. 2.2 – stating that the note "is not primarily focused on modelling " is not a reasonable excuse for adopting a deficient boundary condition. The correct and optimal boundary condition should be used. This is a benchmarking study – so benchmarking the code in question is necessary. 2.2 – The sentence "The initial conditions for the simulation correspond to the" should be changed to describe exactly what parameters were used instead of the descriptive approach to describing the model parameters. 2.2 – What "geotechnical tests" were undertaken to estimate porosity? 2.2 – How was water density measured? 2.2 – I disagree that the unsaturated zone was not relevant. Test

whether it was. Also, it doesn't make sense to say that it was not relevant, but was simulated anyway. 2.2 – It is not adequate to say that the retention represented "fine sand" without giving the retention curve parameters and the retention curve model. 2.2 – There is a tendency in this paper to not give evidence, but to effectively say "trust us" with explanations. The manuscript should be rewritten to remove this. For example, give the Pe equation and the threshold value that was used, and give the Pe that applies to the simulations. What dispersivities were used by others? What has been said about artificial numerical dispersion in SUTRA? 3.1 – This doesn't make sense "The analysis of the water levels in the upstream and downstream tanks confirms that the hydraulic gradient remained constant during the experiment". The hydraulic gradient across the sand tank may have been constant, but gradients within the sand tank may have changed with seawater intrusion. Also, one doesn't analysis the boundary water levels to confirm the gradient, the boundary water levels are controlled so that the gradient across the tank doesn't change.. Table 1 – Give the value for dynamic viscosiry and specific gravity used in getting k from K 3.1 – What is the uncertainty in the Q value of 1.6 L/h? Figure 4 – It seems very unlikely that the experimental Q value was a perfect constant during the grey period (pre-steady-state?). Also, it's unlikely that it was a perfect constant during the period of pumping. Show data points in Figure 4. It is clear from the results in Figure 4 that the experiments were not in steady state when (1) pumping started (i.e., flow is clearly still dropping), and (2) at the end of the experiment (flow is clearly still changing). The results are not steady-state, and the tank was not stabilised before pumping began. This is a significant issue. 3.1 – It doesn't make sense to refer to plus-or-minus two standard deviations, where the standard deviation is described as "of the measures". The flow was changing in time, so the variation in measures has a temporal component. The standard deviation should be the standard deviation about measurements taken for a given condition – i.e., representing experimental error, not the error that arises because the experiment was not run for long enough to establish the conditions (steady-state) that are meant to have arisen. 3.1 – Grammar problems: "Figure 4 also shown the…." And "has an height of $\pm 2$ the stan-
dard deviation". Please proofread carefully the manuscript for these and other issues that were apparent but are not reported in this review. 3.1 – How was EC related to water density? 3.1 – Here it refers to a weak correlation between EC and light absorbance that was not described when this correlation was introduced earlier. Figure 5 and 3.1 – relative to the stained water, the ERT inversion shows a highly variable result. This could have been pre-empted from previous knowledge of ERT. I don't see the benefit of ERT in this research. It doesn't improve on the interpretation of the interface from the food dye. The coefficient of determination indicates a weak correlation – it's value of 0.16 is view over-optimistically in the manuscript. While exploration of this error might be "beyond the scope of this note", I recommend at least speculating on it – especially in regards to experimental error, such as the effect of metallic components of the apparatus. 3.1 – I disagree that the results shows the benefit of ERT – it does not add to the results otherwise obtained. Figure 6 is very difficult to interpret without a better explanation/definition of "density" in the Density plot. This is a continuation of the excessive assumed knowledge and inference that is apparent throughout the manuscript. Figure 7 – problems with this figure include: (a) the black crosses look like a black line. (b) the toe still has a velocity at the end of the experiment – it is therefore not in steady state. 3.2 – The pumping mechanism is the same as a line of closely spaced wells parallel to the coast, or a trench in which groundwater is extracted that is parallel to the coast. This is not a realistic reproduction of a common situation in coastal aquifers. Usually, wells are used at locations that are not placed parallel to the coast and close together. Figure 8 – the slight rise in interface is as apparent in the experimental dye as it is in the ERT. I disagree that this somehow supports the use of ERT here. Generally- the benefit of ERT is very significantly overstated in terms of its contribution to this research. I do not find that it provides any additional benefit to the characterisation of this modelling. Further, I don't see that this case adds significantly to previous sand tank experiments of interface locations, and most certainly does not amount to a new benchmark case that has advantages beyond existing research findings. If anything, the research shows that ERT is not worth using in sand tank experiments of seawater

intrusion if you can see the wedge through the side of the tank. I suggest not using "exactly" when referring to numerical simulations. The outcome of this research is best described by this text "the comparison between observed and simulated data confirm that our experimental facility can be effectively used to develop benchmarks for the validation of density-dependent flow and transport numerical models." That is, the re-search shows that the sand tank can be used to test numerical models. This is not novel – it was known that sand tanks can be used to test numerical models before this research was undertaken.

---

## Editor Comment (EC1) · Brian Berkowitz (Editor) · 15 May 2019

The manuscript – technical note – presents a laboratory facility to investigate processes of saltwater intrusion in coastal aquifers. The aim is to provide benchmark measurements using different techniques as an aid for the testing of numerical models. The detailed review by Referee #1 raises a set of serious concerns, which focus mostly on the lack of novelty and new insights in the manuscript. The referee notes that (i) sand tank models have already been used extensively to provide data for testing of numerical models, and (ii) the current manuscript only confirms that such sand tank models can be used, in principle, in this capacity. The referee also doubts the value

and reliability of the ERT data, as well as detailing other significant criticisms. After careful reading of the manuscript, I must concur with the referee. The manuscript does not offer significant new methodologies or insights. Even with revision to address some of the technical aspects, this major concern will not be addressed. I must therefore, unfortunately, decline publication of this manuscript in HESS.